# A Primary Mediastinal Monophasic Spindle-Cell Synovial Sarcoma with Superior Venacaval Obstruction

**DOI:** 10.3390/diseases10040105

**Published:** 2022-11-17

**Authors:** Deepak Madi, Nikhil Victor Dsouza, Matthew Antony Manoj, Basavaprabhu Achappa, Stergios Boussios

**Affiliations:** 1Department of Internal Medicine, Kasturba Medical College, Mangalore, Manipal Academy of Higher Education, Manipal, 203, Light House Hill Road, Hampankatta, Mangaluru 575001, Karnataka, India; 2Department of Medical Oncology, Medway NHS Foundation Trust, Windmill Road, Gillingham ME7 5NY, UK; 3School of Cancer & Pharmaceutical Sciences, Faculty of Life Sciences & Medicine, King’s College London, London SE1 9RT, UK; 4AELIA Organization, 9th Km Thessaloniki-Thermi, 57001 Thessaloniki, Greece

**Keywords:** mediastinal synovial tumour, synovial cell sarcoma, monophasic spindle-cell sarcoma, superior vena cava obstruction, superior vena cava syndrome

## Abstract

Primary mediastinal sarcoma is a rare tumour that usually presents with nonspecific symptoms such as hoarseness, dyspnoea, and chest pain. Superior vena cava (SVC) syndrome is an extremely uncommon complication that is caused by the compression, invasion, and thrombosis of the SVC or brachiocephalic veins. SVC syndrome can present as asymptomatic cases or as rare life-threatening emergencies with upper airway obstruction and increased intracranial pressure. This report describes the case of a 58-year-old female who presented with swelling of the face, neck, and upper limbs associated with dyspnoea on exertion. The radiological investigations revealed a large well-defined central necrotic peripherally enhancing lesion in the superior mediastinum extending anteriorly with the compression of brachiocephalic veins. A histopathological examination detected spindle cells arranged in fascicles with nuclear atypia with immunohistochemistry positive for creatine kinase (CK), smooth muscle actin (SMA), desmin and CD99. These findings established the diagnosis of a mediastinal monophasic synovial sarcoma with SVC obstruction. The patient was initiated on palliative radiotherapy for the management of the SVC, followed by systemic biological treatment with the tyrosine kinase inhibitor pazopanib, and was clinically improved. It is essential to promptly diagnose and treat this condition, especially when SVC syndrome manifests.

## 1. Introduction

Synovial sarcomas derived its name from the initial assumption that the tumour originated from synovial cells. Soft-tissue sarcomas are rare forms of tumours constituting less than 1% all malignant neoplasms and approximately 10% of all soft-tissue sarcomas. The majority of thoracic tumours are carcinomas, with the prevalence of soft-tissue sarcomas being as low as 0.01% [1].

Recent advances and new research reporting cases of synovial sarcomas arising from structures that do not contain synovial cells have led investigators to believe that synovial sarcomas arise from pluripotent mesenchymal cells capable of divergent differentiation [2]. Synovial sarcomas are commonly reported to affect the younger population and occur in most anatomic sites, including the thoracic cavity. While most of the thoracic cases occur in the pleuropulmonary system, the mediastinum remains an exceptionally rare site of occurrence. A case series of 21 cases, the largest of its kind, reported that the incidence of mediastinal synovial sarcomas was approximately 11.2% of all thoracic synovial sarcomas [3]. The recognition of a specific translocation t(X; 18) (p11; q11) via fluorescence in situ hybridisation (FISH) or reverse transcription polymerase chain reaction (RT-PCR) is a powerful tool in the differential diagnosis with other common neoplasms occurring in the mediastinum, and molecular confirmation remains the gold standard in the diagnosis of these tumours [4]. Due to its aggressive nature, the complete resection of the tumour is essential for a good prognosis. However, as most of the cases are detected during the unresectable stage, external beam radiotherapy is preferred over chemotherapy as the primary choice of treatment.

According to our literature search in the PubMed database using the terms “mediastinal synovial sarcoma” and “primary mediastinal monophasic spindle cell synovial sarcoma”, there were fewer than 30 reported cases. In addition to this, our literature search using the terms “mediastinal monophasic spindle cell synovial sarcoma” and “superior venacaval obstruction” identified only 4 reported cases of that rare clinical coexistence.

Here, we present a case of a 58-year-old female who presented with an occult mediastinal mass causing superior vena cava (SVC) obstruction. The diagnostic work-up demonstrated a monophasic synovial spindle-cell sarcoma—a challenging clinical scenario. In addition to its uncommon occurrence, this case also highlights the importance of recognising rare complications of the disease. Through this case, we aim to increase awareness among doctors and the medical fraternity about the presentation, recognition, complications, and treatment of the disease, especially in the Indian population.

## 2. Case Description

A 58-year-old female with prior comorbidities of hypertension, diabetes mellitus, and hypothyroidism presented to the hospital with exertional dyspnoea over a month, hoarseness of voice, and oedematous face and neck over 15 days. Otherwise, she denied a prior history of haemoptysis, dizziness, smoking or tuberculosis.

On preliminary examination, the patient was conscious with tachypnoea and tachycardia. She had puffiness with the flushing of the face, periorbital region, and neck. Oedema of both upper limbs was also noted. Jugular venous pressure was elevated, and the rest of the systemic examination was unremarkable. Preliminary blood reports were within normal limits, and a chest X-ray showed a large mass with respect to the mediastinal compartment obscuring the bilateral hila and arch of the aorta (Figure 1).

The performed contrast-enhanced computerised tomography (CECT) of the chest revealed a large, well-defined, centrally necrotic, peripherally enhancing mass lesion in the superior mediastinum extending to the anterior mediastinum. There was displacement of the trachea to the left side, compression of brachiocephalic veins and proximal SVC, and a possibility of sclerotic skeletal metastasis in the vertebrae (Figure 2). A CT-guided biopsy was planned, and a fluorodeoxyglucose (FDG) positron emission tomography (PET) scan (SUV-13.1) was performed to differentiate between a malignancy or a mass with an infective lesion and to look for metastasis. The scan revealed a large 10.7 × 6.5 × 6.2 cm well-defined mass lesion of neoplastic aetiology with a mass effect—likely of thymic origin, but with no evidence of metastasis (Figure 3).

In histopathology, the CT-guided biopsy of the lesion exhibited a core of tumour tissue composed of spindle cells arranged in fascicles with nuclear atypia (Figure 4A) with the possibility of a thymic or mesenchymal spindle-cell neoplasm. Immunophenotyping was positive for creatine kinase (CK), smooth muscle actin (SMA), desmin, epithelial membrane antigen (EMA) and CD99 (Figure 4B) with MiB 42%. It was negative for p63, S100, and CD117, with a final impression of features favouring a mesenchymal malignant neoplasm suggestive of monophasic synovial spindle-cell sarcoma.

The patient was referred to the multidisciplinary team of medical, surgical, and radiation oncologists in view of the lesion being unresectable as a result of the large vessel involvement. As chemotherapy is considered to be less effective in monophasic variants, it was decided to treat the patient with palliative radiotherapy. During her admission, she underwent radiotherapy that resolved the SVC obstruction. The patient is currently on systemic targeted treatment with the tyrosine kinase inhibitor pazopanib. The patient has since then been on regular follow-up. However, due to logistical reasons during the COVID-19 pandemic, the patient is currently following up at a nearby center and faring well. The patient will be undergoing an interval PET/CT scan to look for treatment response/disease progression.

## 3. Discussion

Due to its appearance being similar to that of synovial tissues under light microscopy, tumours arising near the joints were earlier termed as synovial sarcoma. However, over time, it was found that it originated from pluripotent mesenchymal cells near joint surfaces, tendons, tendon sheaths, fascial aponeurosis, and juxta articular membranes. Synovial sarcomas comprise up to 5–10% of all soft tissue sarcomas. The age of presentation is between the second and fourth decades. The common sites of occurrence are in the lower extremities and near the joints [5]. They occur at other sites of the body, but only a few cases reported the mediastinum as a site of origin [2,6]. However, in our case, the age of presentation was the fifth decade, and the site of origin was the mediastinum, which was unusual.

Morphologically, due to various growth patterns in synovial sarcoma, it was divided into 3 distinct subtypes: monophasic, biphasic, and poorly differentiated. The classically recognized variant is the biphasic type, which consists of a mixture of epithelial and bland-looking spindle-shaped cells. On the other hand, the monophasic type exhibits either spindle cells or epithelial components only [5,6].

The biphasic subtype can be easily diagnosed by histopathology; however, the poorly differentiated and monophasic subtypes can be misdiagnosed as a fibrosarcoma, leiomyosarcoma, malignant peripheral nerve sheath tumour, or hemangiopericytoma. Hence, the role of immunohistochemistry is paramount in confirming the diagnosis of the patient. This was observed in our case, as the clinical features, radiology, and histopathology could not confirm the diagnosis of the disease [6].

Immunohistochemistry helps in excluding two conditions that closely resemble monophasic synovial sarcoma, namely, malignant peripheral nerve sheath tumour (S100-positive) and solitary fibrous tumour (CD34-positive). Classically synovial sarcomas are positive for cytokeratin, CD56, EMA, CD99, and Bcl-2c [5,7]. In the cases of primary mediastinal and pulmonary synovial sarcomas, Bcl-2 and CD56 are very sensitive markers [8]. In the modern era, the availability of molecular testing, namely, polymerase chain reaction or FISH, has improved the diagnostic specificity of synovial sarcomas due to its unique translocation t(X;18) (p11.2; q11.2) between chromosomes X and 18 [2].

There are few reported cases of patients with synovial sarcoma presenting with thoracic outlet obstruction. According to the literature, the patients most often present with shoulder pain, chest pain, and/or shortness of breath [9]. However, the patient in our report originally developed features suggestive of SVC obstruction.

Due to the fact that synovial sarcomas are highly aggressive in nature, prognostic factors play a vital role in the therapeutic approach of these patients, namely, tumour size (>5 cm), distant metastasis, resection margin status, and the degree of histological differentiation. When the tumour is resectable, surgery is the primary choice of treatment with the aim to achieve negative surgical margins. However, when the tumour is close to critical structures, complete resection is not possible; hence, neoadjuvant chemotherapy and/or radiotherapy is preferred [10]. In our case, due to the size and encasement of the large vessels, the tumour was inoperable, and the patient underwent palliative radiotherapy to relieve the SVC obstruction and to consider the possibility of surgery at a later date. Radiotherapy and chemotherapy are effective in the treatment of unresectable nonmetastatic tumours as they are highly responsive to the same. A meta-analysis reported that the 5-year overall survival was 35.7% in patients with unresected large tumours and advanced diseases versus 63% in patients who underwent complete resection of the tumour [7].

Though there is no rule of thumb when it comes to the treatment of mediastinal synovial sarcomas, the literature states that surgery followed by adjuvant multimodality therapy is the best course of treatment [11]. In general, anthracycline-based chemotherapy is the first-line treatment for advanced soft-tissue sarcomas. Ifosfamide also has well-documented efficacy in synovial sarcoma in the palliative setting. The combination of doxorubicin and ifosfamide may produce better outcomes in advanced disease than those of other chemotherapy regimens. A high dose of ifosfamide may also be considered, but its efficacy is still investigated. An alternative treatment is the combination of gemcitabine and docetaxel, especially in patients with poor tolerance or resistance to doxorubicin and ifosfamide. Other chemotherapeutics, such as trabectedin, eribulin gemcitabine/docetaxel, or dacarbazine, have been investigated and induced responses. Angiogenesis is crucial in tumour progression; hence, antiangiogenic agents may be a reasonable treatment in soft-tissue sarcomas. They can either be multiple tyrosine kinase inhibitors, such as pazopanib, sorafenib, sunitinib, and cediranib, or monoclonal antibodies such as bevacizumab. Pazopanib was investigated in patients with advanced soft-tissue sarcomas and demonstrated improved progression free survival in a Phase III trial [12]. Our team decided to treat the patient with pazopanib for two main reasons. First, chemotherapy is less effective in monophasic variants, and second, the overexpression of platelet-derived growth factor receptor antigen in synovial sarcomas renders tyrosine kinase inhibitors most likely to be effective.

Up to now, 4 cases of patients who developed SVC obstruction secondary to primary mediastinal monophasic spindle-cell synovial sarcoma have been reported in the English literature. Table 1 summarises these reported in the English literature.

## 4. Conclusions

Mediastinal monophasic spindle-cell sarcoma with SVC syndrome is a rare tumour with a rare complication. The prompt diagnosis and early treatment of this life-threatening complication are crucial. We emphasise that clinicians should keep in mind synovial sarcomas as differential diagnosis, even though the mediastinum is an unusual site. The diagnosis must be confirmed with the aid of histopathology and immunohistochemical markers.

## Figures and Tables

**Figure 1 diseases-10-00105-f001:**
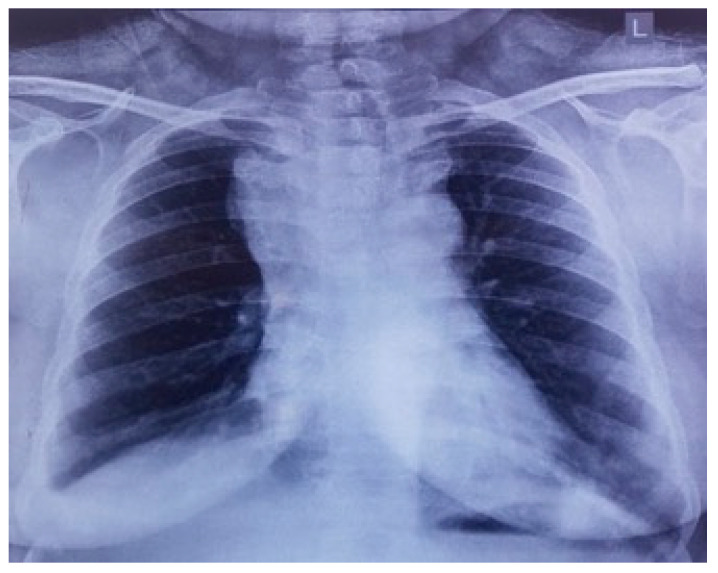
Chest X-ray showing widened superior mediastinum.

**Figure 2 diseases-10-00105-f002:**
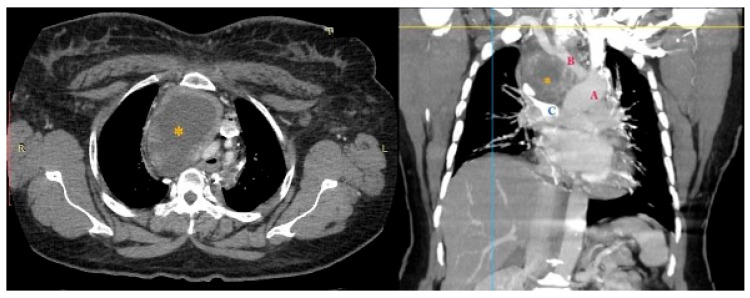
CECT showing large central necrotic lesion measuring 8.4 × 6.1 × 8.2 cm involving the superior mediastinum, displacing the trachea to the left, and compressing the brachiocephalic veins and the proximal SVC. * necrotic lesion.

**Figure 3 diseases-10-00105-f003:**
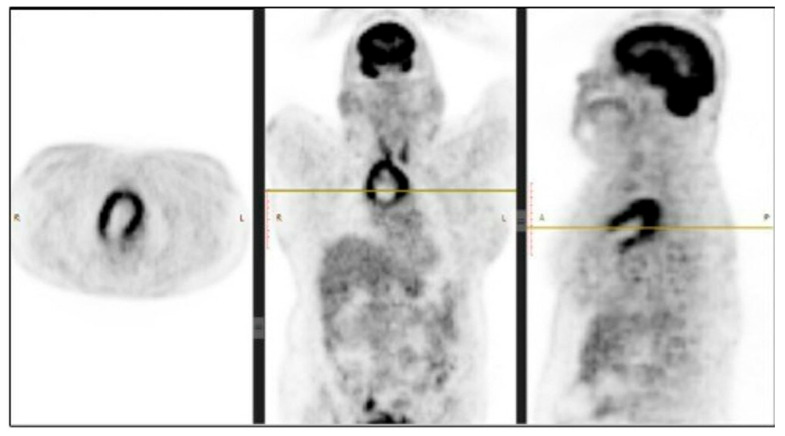
PET scan showing metabolic activity in the mediastinal mass. Abbreviations: R: Right; L: Left.

**Figure 4 diseases-10-00105-f004:**
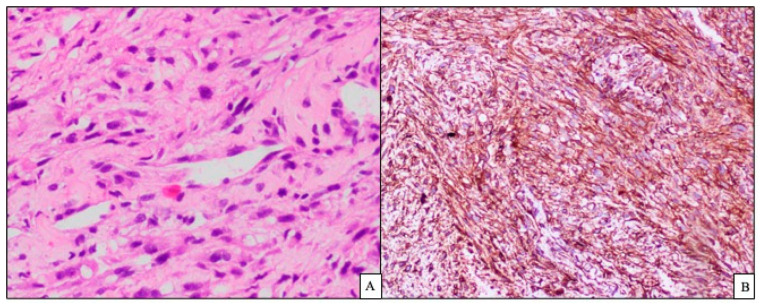
Histopathology showing spindle cells arranged in fascicles with (**A**) nuclear atypia 20× and (**B**) immunohistochemistry CD 99-positive.

**Table 1 diseases-10-00105-t001:** Total reported cases of patients with mediastinal monophasic spindle-cell synovial sarcoma who developed SVC obstruction.

Patient No.	Age/Gender	Presentation	Treatment	Outcome	Ref.
1	35/M	Dyspnoea, facial oedema, voice hoarseness, and engorged neck veins	Diuretics, dexamethasone, oxygen inhalation, and 3 cycles of chemo (ifosfamide 2400 mg/m^2^ (Days 1–5) and doxorubicin 37.5 mg/m^2^ (Days 1 and 2))	Clinical improvement On active surveillance	[6]
2	27/M	Dyspnoea and chest discomfort aggravated on exertion	3 cycles of fosfamide and adriamysin followed by gemcitabine and docetaxel due to inoperable disease	Clinical improvement On follow up	[11]
3	44/M	Shortness of breath, facial and upper extremity oedema	Tracheal stent placement, chemo	Death	[13]
4	14/F	Dyspnoea on exertion, facial and neck oedema	Resuscitation Decompressive RT was planned	Death	[14]

Abbreviations: SVC: superior vena cava; Ref.: reference; M: male; F: female; chemo: chemotherapy; RT: radiotherapy.

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
