# Peer review of "A Primary Mediastinal Monophasic Spindle-Cell Synovial Sarcoma with Superior Venacaval Obstruction"

_diseases, 2022, doi:10.3390/diseases10040105_

Round 1

Reviewer 1 Report

This is a straightforward case report on a patient with a monophasic synovial sarcoma of the mediastinum that caused obstruction of the superior vena cava.

The diagnosis is well established by means of immunohistochemistry. No need for demonstration of the characteristic translocation t(X;18) (p11.2; q11.2) in this case, in view of the clear-cut immunohistochemistry.

In view of the localization of the tumor in the area of the thymus, it might have been discussed whether the sarcoma could have developed as a somatic type malignancy in a mediastinal type II germ cell tumor. This can practically be ruled out by the age of the patient (58 years of age) and by the female sex of the patient, as mediastinal type II germ cell tumors occur predominantly in adolescent males.

There are a couple of suggestions for improvement of the text, which are shown in the accompanying attachment.

Author Response

Thank you for your comments. We have incorporated all the changes into the article as suggested by you for the improvement of the literature text.

  1. Please indicate the metastasis in the vertebrae in figure 2.

Thank you for the comment. The CT mentioned the possibility of sclerotic skeletal metastasis in the vertebrae but the PET CT confirmed no metastasis. Furthermore, the possible metastatic lesion on the CT is not visualized on the vertebrae presented in the manuscript.

Reviewer 2 Report

Synovial sarcoma of the mediastinum is a rare neoplasm that has overlapping histological and immunophenotypic features with other tumours in the differential diagnosis. This article describes a case of such disease. There are some points that need to be clarified.

1.  In the introduction, there was no mention of any significant highlight. For an article, rarity alone cannot be the highlight to write, there must be other more valuable reasons to contribute to the literature. To answer the question, "What is the article's value?”

2.  In the methods used for diagnosis, the author sits in judgment that the lesion is a malignancy relies on PET/CT, while it is usually hard to differentiate malignancy from inflammatory disease on PET/CT.  Another defect Is that the SUV value of PET/CT did not present in the article results.

3.  There is a big question and unclearness in the exposition of treatment progress, as the authors only mention a few words about the use of targeted therapy without declaring detailed use for this treatment.

4. In terms of Synovial sarcoma treatment, according to NCCN guidelines it is reasonable to use chemotherapy as first-line treatment first. But the authors has directly targeted therapy, which is a big question mark that might cause confusion for the readers if not clarified.

5. Duration of follow-up and patient outcome?

6.  Finally, in the discussion section, discuss more about emergent treatment for the “life threatening condition” as the authors mentioned in his Conclusion.

Author Response

  1. In the introduction, there was no mention of any significant highlight. For an article, rarity alone cannot be the highlight to write, there must be other more valuable reasons to contribute to the literature. To answer the question, "What is the article's value?”

Thank you for your comment. We have addressed this comment in the last paragraph of the introduction.

In addition to its uncommon occurrence, this case also highlights the importance of recognizing rare complications of the disease. Through this case, we aim to increase awareness among the doctors and the medical fraternity about its presentation, recognition, complications and treatment of the disease especially in the Indian population.”

  1. In the methods used for diagnosis, the author sits in judgment that the lesion is a malignancy relies on PET/CT, while it is usually hard to differentiate malignancy from inflammatory disease on PET/CT.  Another defect Is that the SUV value of PET/CT did not present in the article results.

Thank you for your comment. We would like to clarify that, we initially did a CT, which enabled us to consider a differential diagnosis. The patient was then taken up for a CT guided biopsy, following which, A PET scan was done as advised by the MDT (Multidisciplinary team). The SUV value has been added into the article.

A CT-guided biopsy was planned and a fluorodeoxyglucose (FDG) positron emission tomography (PET) scan (SUV-13.1) was done to differentiate a malignancy or a mass with an infective lesion and to look for metastasis.”

  1. There is a big question and unclearness in the exposition of treatment progress, as the authors only mention a few words about the use of targeted therapy without declaring detailed use for this treatment.

  2. In terms of Synovial sarcoma treatment, according to NCCN guidelines it is reasonable to use chemotherapy as first-line treatment first. But the authors has directly targeted therapy, which is a big question mark that might cause confusion for the readers if not clarified.

Thank you for the above comments. The response to both the comments have been explained in the last but one paragraph in the discussion.

In general, anthracycline-based chemotherapy comprise first-line treatment for advanced soft tissue sarcomas. Ifosfamide has also well documented efficacy in synovial sarcoma in the palliative setting. The combination of doxorubicin and ifosfamide may produce better outcomes in advanced disease than other chemotherapy regimens. High-dose ifosfamide may also be considered, but its efficacy is still investigated. An alternative treatment is the combination of gemcitabine and docetaxel, especially in patients with poor tolerance or with resistance to doxorubicin and ifosfamide. Other chemotherapeutics, such as trabectedin, eribulin gemcitabine/docetaxel or dacarbazine have been investigated and induced responses. Angiogenesis is crucial in tumor progression, and hence anti-angiogenic agents may be a reasonable treatment in soft tissue sarcomas. They can either be multiple tyrosine kinase inhibitors, such as pazopanib, sorafenib, sunitinib, and cediranib, or monoclonal antibodies such as bevacizumab. Pazopanib has been investigated in patients with advanced soft tissue sarcomas and has demonstrated improved progression free survival in a phase III trial. Our team decided to treat the patient with pazopanib for two main reasons. Firstly, chemotherapy is less effective in monophasic variant and secondly, the overexpression of platelet-derived growth factor receptor antigen in synovial sarcomas makes tyrosine kinase inhibitors most likely to be effective.”

  1. Duration of follow-up and patient outcome?

Thank you for your comment. We have addressed this in the last paragraph of the case report section of the article.

The patient has since then been on regular follow-up. However, due to logistic reasons during the COVID-19 pandemic, the patient is currently following up at a nearby center and is presently continuing treatment.”

6.  Finally, in the discussion section, discuss more about emergent treatment for the “life threatening condition” as the authors mentioned in his Conclusion.

Thank you for your comment. We have addressed this in our discussion section, 2nd last paragraph.

However, when the tumour is close to critical structures, complete resection is not possible and hence, neoadjuvant chemotherapy and or radiotherapy is preferred. In our case, due to the size and encasement of the large vessels, the tumour was inoperable and the patient underwent palliative radiotherapy to relieve the SVC obstruction and to consider the possibility of surgery at a later date. Radiotherapy and chemotherapy are effective in the treatment of unresectable non-metastatic tumours as they are highly responsive to the same. A meta-analysis reported that the 5-year overall survival was 35.7% in patients with unresected large tumours and advanced diseases, versus 63% in patients who underwent complete resection of the tumour.”

Reviewer 3 Report

Dear Authors, 

thank you very much for the interesting case report. Even though 4 similar cases have already been described in the literature, this particular presentation of medialstinal synovial sarcoma is again interesting for the reader. 

I would recommend the publication of the case report after a small modification. In my opinion, the database in which you performed the literature search should be mentioned. 

Yours sincerely 

Author Response

  1. I would recommend the publication of the case report after a small modification. In my opinion, the database in which you performed the literature search should be mentioned. 

Thank you for comment. We have addressed this in the introduction section of the article, 2nd last paragraph.

According to our literature search in the PubMed database using the terms “mediastinal synovial sarcoma”, “primary mediastinal monophasic spindle cell synovial sarcoma”, there were less than 30 reported cases.”

Round 2

Reviewer 2 Report

  1.  Duration of follow-up and patient outcome?

    Thank you for your comment. We have addressed this in the last paragraph of the case report section of the article.

    The patient has since then been on regular follow-up. However, due to logistic reasons during the COVID-19 pandemic, the patient is currently following up at a nearby center and is presently continuing treatment.”

    The story is incomplete. Whether a treatment is appropriate should be tested by followup and outcome. I would still suggest the authors to add some more details regarding duration of followup and outcome. You may obtain this information by phone, or you should at least report some patient reported outcome instead.

Author Response

Thank you for your comment.

Please, refer to the final paragraph of the case description.

"The patient has since then been on regular follow-up. However, due to logistic reasons during the COVID-19 pandemic, the patient is currently following up at a nearby center and is currently doing well. The patient will be undergoing an interval PET/CT scan to look for treatment response/disease progression."